# Investigating Nanoindentation Creep Behavior of Pulsed-TIG Welded Inconel 718 and Commercially Pure Titanium Using a Vanadium Interlayer

**Tauheed Shehbaz [1], Fahd Nawaz Khan [1], Massab Junaid [2] and Julfikar Haider [3],***

[1] Faculty of Materials and Chemical Engineering, Department of Materials Science, Ghulam Ishaq Khan Institute of Engineering Sciences and Technology, Topi 23640, Pakistan; tauheed@giki.edu.pk (T.S.); fahd@giki.edu.pk (F.N.K.)

[2] Faculty of Mechanical Engineering, Ghulam Ishaq Khan Institute of Engineering Sciences and Technology, Topi 23640, Pakistan; massab@giki.edu.pk

[3] Advanced Materials and Surface Engineering (AMSE) Research Centre, Manchester Metropolitan University, Chester Street, Manchester M1 5GD, UK

\* Correspondence: j.haider@mmu.ac.uk

**Abstract:** In a dissimilar welded joint between Ni base alloys and titanium, creep failure is a potential concern as it could threaten to undermine the integrity of the joint. In this research, the mechanical heterogeneity of a Pulsed TIG welded joint between commercially pure titanium (CpTi) and Inconel 718 (IN718) with a vanadium (V) interlayer was studied through a nanoindentation technique with respect to hardness, elastic modulus, and ambient temperature creep deformation across all regions (fusion zones and interfaces, mainly composed of a dendritic morphology). According to the experimental results, a nanohardness of approximately 10 GPa was observed at the V/IN718 interface, which was almost 70% higher than that at the V/CpTi interface. This happened due to the formation of intermetallic compounds (IMCs) (e.g., $Ti_2Ni$, $NiV_3$, $NiTi$) and a (Ti, V) solid solution at the V/IN718 and V/CpTi interfaces, respectively. In addition, nanohardness at the V/IN718 interface was inhomogeneous as compared to that at the V/CpTi interface. Creep deformation behavior at the IN718 side was relatively higher than that at different regions on the CpTi side. The decreased plastic deformation or creep effect of the IMCs could be attributed to their higher hardness value. Compared to the base metals (CpTi and IN718), the IMCs exhibited a strain hardening effect. The calculated values of the creep stress exponent were found in the range of 1.51–3.52 and 2.52–4.15 in the V/CpTi and V/IN718 interfaces, respectively. Furthermore, the results indicated that the creep mechanism could have been due to diffusional creep and dislocation climb.

**Keywords:** P-TIG welding; CpTi; Inconel 718; vanadium interlayer; nanoindentation; creep

## 1. Introduction

The current trend in industry is toward making multifunctional components by integrating several similar and dissimilar materials. The welding and joining of dissimilar alloys is a crucial procedure in the production of hybrid structures, and a number of researchers have been prompted to look at the welding of various dissimilar alloy combinations [1–5]. Superior mechanical properties and oxidation resistance at high temperatures have made Inconel 718 (IN718), a nickel-based superalloy, a useful material with a range of aerospace applications, such as in aeroengine and gas turbine design [6,7]. Along with IN718, due to its higher strength and excellent corrosion resistance, commercially pure titanium (CpTi) has been extensively used in the airframes of aircrafts and in the petrochemical industry. Due to these special properties, titanium- and nickel (Ti/Ni)-based dissimilar alloy joints can afford significant advantages in many industries [8–10]. For instance, in designing the compressor section of aero engines, one strategy to reduce the net weight of the engine is to use a transition joint between the titanium alloys and the nickel-based

super alloys [11]. These joints, making efficient use of both alloy properties, can improve the performance of equipment, save on material cost, and improve design. Similarly, the use of Inconel and titanium alloy joints which have the same galvanic potential offers the best possible combination when it comes to resisting bimetallic corrosion in structures used in marine contexts [12]. However, in general, a problem arises when the direct joining of the two alloys is compromised by the formation of brittle Ni/Ti IMCs [13–17]. This problem can be overcome, however, by the choice of a proper welding process, input process parameters, and the insertion of a suitable metallic interlayer between the base alloys [18–20].

Dissimilar alloy joints often fail during service. These failures can be due to a number of reasons, such as the differences in microstructural and mechanical properties across the weld joint, and the fact that the coefficient of thermal expansion (CTE) of the two alloys results in creep at the interface (the CTE values for CpTi and IN718 are typically $8.6 \times 10^{-6}$/K and $13 \times 10^{-6}$/K, respectively) [21]. Furthermore, the presence of large residual stresses due to differences in chemical composition, thermal conductivity, melting point, as well as metallurgical incompatibility, can all affect the creep performance of the joints [22]. A high creep strength is essential to ensuring that a structure performs correctly, because if creep deformation exceeds the critical failure deformation specified in the design, the part ceases to operate safely [23]. Therefore, it is critical to investigate creep deformation behavior in dissimilar welded joints.

Creep is the time-dependent deformation under a constant load or stress less than the material's yield strength at constant temperature. Li et al. [24] reported that during nanoindentation, indentation creep occurs between room temperature to half of that material's melting point (0.5 Tm).

The nano-indentation experiment is one of the most widely used non-destructive test procedures for determining the mechanical properties of bulk materials, coatings, thin films, and other materials. One of the interesting applications of the nanoindentation technique is indentation creep testing [25]. In comparison to macroscopic creep deformation testing, which utilizes typical uniaxial compression or tension, the nanoindentation creep test has been regarded as a time-saving, reliable, convenient, easy, and non-destructive approach to investigating the micro- and nanomechanical properties of diverse materials [26,27]. The creep properties of a relatively tiny volume of material in the immediate vicinity of the indenter tip are studied in nanoindentation tests. The loading rate during indentation is two to three orders of magnitude faster, and the stress under the indenter is substantially more convoluted than in standard creep tests [24].

During welding, mechanical strength is associated with a variety of microstructures evolved in different metallurgical/welding zones. The nanoindentation technique has been used to investigate the creep strength of weldments because the microstructural gradient is much more pronounced in welded alloys. The rate-jump approach was utilized by Nguyen et al. [28,29] to study the strain rate sensitivity of SM 490 and SS 400 structural steel welded joints using nano indentation with a low-cycle fatigue loading. The authors concluded that with increasing strain rate indentation, both yield strength and indentation hardness tend to increase. Gao et al. [30] investigated the creep behavior of a P92 steel weld joint after creep–fatigue loading using the nanoindentation technique and reported that the fine-grained heat-affected zone had lower hardness and creep resistance than the other zones of the weld joint. Song et al. [31] also applied the nanoindentation technique to study the creep deformation in different zones of SA508Gr3 steel-welded joints. However, nanoindentation has not been used to analyze titanium and nickel-based alloys weldments.

The use of nanoindentation on the CpTi, IN718 and vanadium interlayer weld joint is employed in the present work to demonstrate the complicated metallurgical reactions in the narrow regions proximal to the fusion boundary. Nanohardness line scans, nanohardness maps, and the ambient temperature creep behavior of these regions were measured using nanoindentation.

## 2. Materials and Methods

In the present study, TIG welding method was adopted to join 80 mm × 50 mm × 1 mm sheets of CpTi and IN718. A vanadium (V) foil with an approximate thickness of 350 μm was placed as an interlayer between the CpTi and IN718 plate edges. The chemical composition of the alloys is given in Table 1. The samples were welded in butt configuration and the tungsten electrode was positioned in the middle of the vanadium interlayer, as shown in Figure 1.

**Table 1.** Chemical composition (at.%) of IN718, CpTi, and V as measured by EDS.

| Elements | IN718 | CpTi | V |
|---|---|---|---|
| Al | 0.88 | - | - |
| Ti | 1.02 | Bal. | - |
| Cr | 21.75 | - | - |
| Fe | 18.61 | 0.15 | - |
| V | - | - | 100 |
| Ni | 50.75 | - | - |
| Nb | 5.29 | - | - |
| Mo | 1.61 | - | - |
| Co | 0.09 | - | - |
| C | - | 0.04 | - |
| O | - | 0.15 | - |

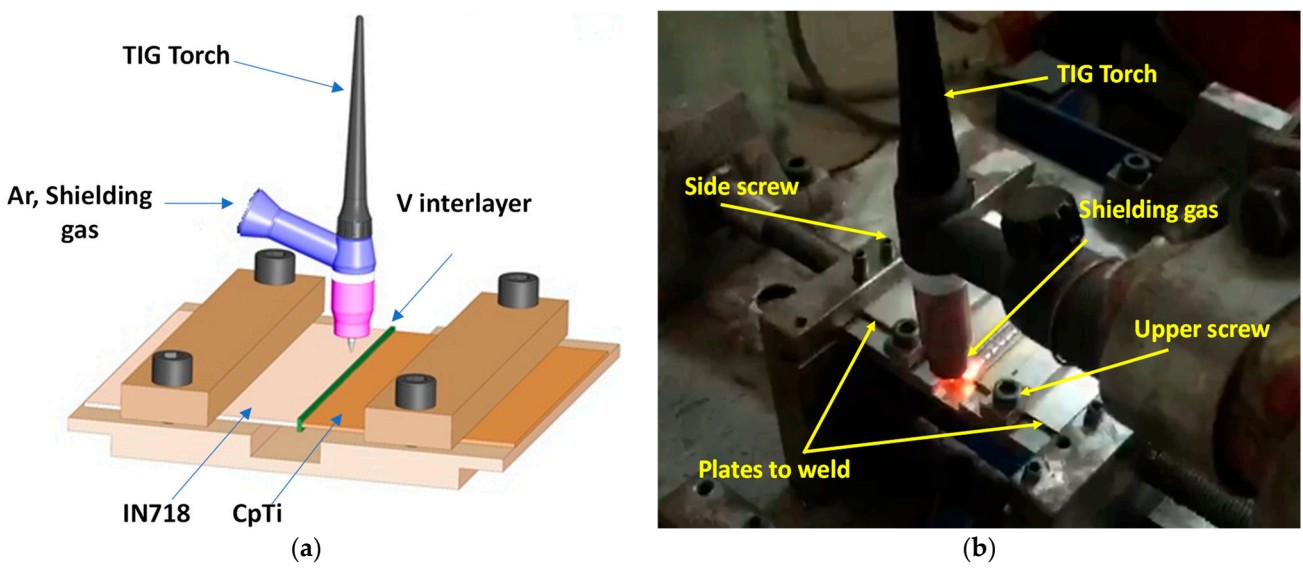

**Figure 1.** (**a**) Schematic diagram. (**b**) Experimental set-up of TIG welding process.

To avoid any contamination in the welded joints, all the sheets were shielded by argon gas during the welding process. After preliminary trials using various welding parameters, the following optimized process parameters were employed in the present study: primary current, 47 A; background current, 84 A; voltage, 11 V; Arc length, 3 mm; and welding speed, 450 mm/min.

### 2.1. Microstructural Caharacterization

The specimens for microstructural and creep characterizations were sectioned from the welded plate using the electric discharge machining (EDM) technique. After cold mounting, the samples were ground using silicon carbide (SiC) papers up to 4000 grit size. The final polishing was performed with a micro cloth using 0.25 μm diamond paste. The etchants used for CpTi, V, and IN718 were Kroll solution (2% HF in distilled water), 30 mL glycerin + 10 mL HF + 10 mL $HNO_3$ and 2g $CuCl_2$ + 15 mL HCl + 5 mL $HNO_3$ +

5 mL $H_2O$, respectively. Microstructural examination of the dissimilar joint was carried out using scanning electron microscopy (SEM) (LYRA3, TESCAN, Brno, Czech Republic) with an energy dispersive spectroscopy (EDS) attachment. The X-ray diffractometer (Philips, PW 3710 Eindhoven, The Netherlands) was operating at 25 mA and 40 kV, using a CuK$\alpha$ ($\lambda$ = 0.1542 nm) radiation source. The diffraction patterns were collected over an angular range of 2$\theta$ = 15° to 80°, and a step size of 0.05° per step per sec was used to investigate the phases present within the joint microstructure.

### 2.2. Nanoindentation

Nanoindentation was conducted at ambient temperature using a nanoindenter (iMicro, Nanomechanics, Oak Ridge, TN, USA). During the nanoindentation test, a three-sided Berkovich tip at a maximum load of 200 mN in a load control mode was used to measure the hardness and elastic modulus at different regions of the dissimilar joint. Each weld zone was marked with a 2 × 2 grid. In order to avoid indentation impacts on each other, the indents were separated by 40 μm. Both nanohardness line scans and nanohardness maps were produced along the interfaces and zones created during welding.

For creep tests, the maximum load for nanoindentation was kept at approximately 200 mN, with a loading strain rate of 0.1 s$^{-1}$. After reaching the maximum load limit, the indenter was kept for 200 s to observe the variation in creep depth over time. Following that, the indenter was instantly unloaded to 10% of its maximum load and kept for 40 s to monitor the drift velocity. Finally, the indenter was fully unloaded from the specimen surface to complete the test.

### 2.3. Calculation Methods

### 2.3.1. Hardness and Elastic Modulus

Hardness (*H*) and elastic modulus (*E*) values were calculated by the Oliver and Phar method [32]. The hardness value was determined from load (*P*) and Projected area (*Ac*) of the indentation using Equation (1):

$$H = \frac{P}{Ac} \tag{1}$$

The reduced modulus of a material can be calculated from the elastic and plastic deformation observed in the load vs. depth curves using Equation (2). In addition, the modulus can also be calculated from the slope of the upper region of the unloading curve of the *P–H* plot. In our case, the elastic modulus is calculated directly from nanoindentation according to the Oliver and Phar method.

$$\frac{1}{E_r} = \left[ \frac{1 - v_i^2}{E_i} + \frac{1 - v_s^2}{E_s} \right] \tag{2}$$

### 2.3.2. Indentation Strain Rate

The experimental data from the holding stage at 200 s was used to study the creep behavior in nanoindentation testing. The indentation strain rate for the Berkovich indenter is stated by Poisl et al. [33] as:

$$\dot{\varepsilon} = \frac{\dot{h}}{h} = \frac{1}{h}\frac{dh}{dt} \tag{3}$$

where $\dot{\varepsilon}$ and $\dot{h}$ are creep strain rate and rate of indentation depth, respectively. During the holding phase, the indentation depth rate data was retrieved and fitted using the empirical model [34]. This creep model is represented by Equation (4):

$$h = h_i + a(t - t_i)^{\frac{1}{2}} + b(t - t_i)^{\frac{1}{4}} + c(t - t_i)^{\frac{1}{8}} \tag{4}$$

where *h* is depth of penetration of indenter, *t* is creep time, and *a*, *b*, *c*, $h_i$, and $t_i$ are the best fit parameters for the equation based on the creep time curve.

### 2.3.3. Indentation Stress

A relationship for the creep stress in the holding stage can be simply calculated [27–35] using Equation (5):

$$\sigma = \left(\frac{H}{3}\right)\left(\frac{h_{max}}{h}\right)^2 \tag{5}$$

where $\sigma$ is stress, $H$ is nanohardness, $h$ is instantaneous depth, and $h_{max}$ is maximum loading depth.

### 2.3.4. Creep Stress Exponent

The creep stress exponent is widely used to explain the stress dependency of the deformation rate throughout the secondary stage of the process, as well as the steady-state creep stage. According to the literature on the subject [36], room temperature creep behavior can be studied by utilizing a power law relationship between stress and creep strain rate. The following equation is an expression of the power law relationship:

$$\dot{\epsilon} = k\sigma^n \tag{6}$$

where $k$ is a material constant, and $n$ is the creep stress exponent. The exponent $n$ is calculated as an indicator for identifying the primary creep mechanism by measuring the slope of the $\ln(\dot{\epsilon})$ vs $\ln(\sigma)$ plot provided in Equation (7) under isothermal conditions:

$$n = \partial \ln(\dot{\epsilon}) / \partial \ln(\sigma) \tag{7}$$

## 3. Results and Discussion

### 3.1. Microstructural Characterization

A complete cross-sectional view of microstructural details of the welded joint is shown in Figure 2a. The TIG arc facing side depicts the common melt zone (the green dotted region) where the melting and dissolution of the CpTi and IN718 plates with the V foil is shown by an EDS line scan. The mixing of Ti, Ni and V triggered several metallurgical reactions which led to the formation of intermetallic compounds (IMCs). The fusion zone (FZ1) on the IN718 side shown in Figure 2b is comprised of relatively coarse dendrites.

Based on the EDS analysis (Table 2) and binary phase diagrams for Ti/Ni and Ni/V [37], the major phases in this region are TiNi, $Ti_2Ni$ and titanium nitride (TiN). The presence of TiN can be attributed to the absorption of atmospheric nitrogen at the surface of the melted zone. Chatterjee et al. [17] reported similar results in the laser welding of pure nickel with commercially pure titanium.

**Table 2.** EDS (at.%) analysis of the CPTi/V/Inconel 718 joint. The position of the EDS analysis indicated in Figure 2a–c.

| Spot | V | Fe | Ni | Cr | Ti | Nb | N | Possible Phase |
|------|-------|------|-------|-------|-------|------|-------|----------------|
| 1 | 16.14 | 7.16 | 30.27 | 2.98 | 43.44 | | | $Ti_2Ni$ |
| 2 | 5.34 | 1.92 | 8.69 | 0.94 | 56.44 | 1.62 | 25.06 | TiN |
| 3 | 27.51 | 4.79 | 36.97 | 1.73 | 23.15 | 5.85 | | NiTi |
| 4 | 31.22 | 3.65 | 8.17 | 3.87 | 53.10 | | | SS (Ti, V) BCC Phase |
| 5 | 69.57 | 6.98 | 8.82 | 10.37 | 4.26 | | | $NiV_3$ |

Figure 2c shows the melt zone right above the V interlayer, composed of a dendritic morphology, dispersed in the (Ti, V) solid solution. The EDS results in Table 2 revealed a considerably greater amount of Ni and Ti in these dendrites, possibly leading to the formation of $Ti_2Ni$ in the V and Ti solid solution. Furthermore, Figure 2d shows the FZ3 zone with a columnar dendritic morphology and the presence of $Ni_3V$ and NiTi in the solid solution of Ti and V.

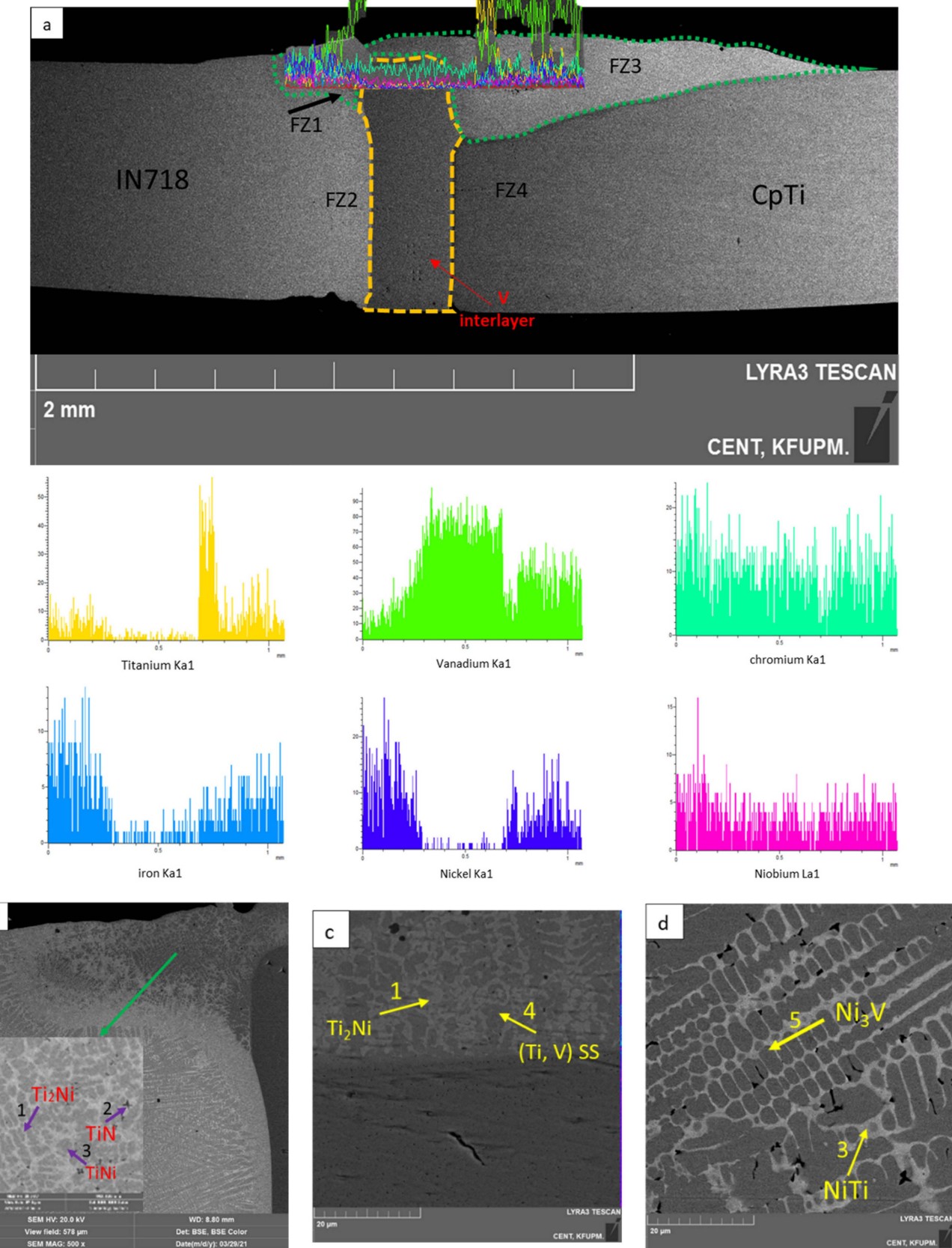

**Figure 2.** (**a**) SEM micrographs of the CpTi/V/IN718 joint with EDS line scan. Microstructures in the common melt zone of all three materials in the arc facing side: (**b**) Inconel 718 side; (**c**) region right above the V interlayer; and (**d**) the CpTi side.

Figure 3a shows the columnar microstructure (FZ2 zone) of IN718 separated from the V interlayer with the elemental distribution of V, Ni, Fe and Cr at the interface. It appears from the distribution that V diffuses significantly in IN718, whereas Ni contents in the vanadium are relatively lower than that in IN718. This shows that neither element exhibited mutual diffusion. The creation of brittle $NiV_3$ (brittle IMCs) resulted from the diffusion of V in Ni [38].

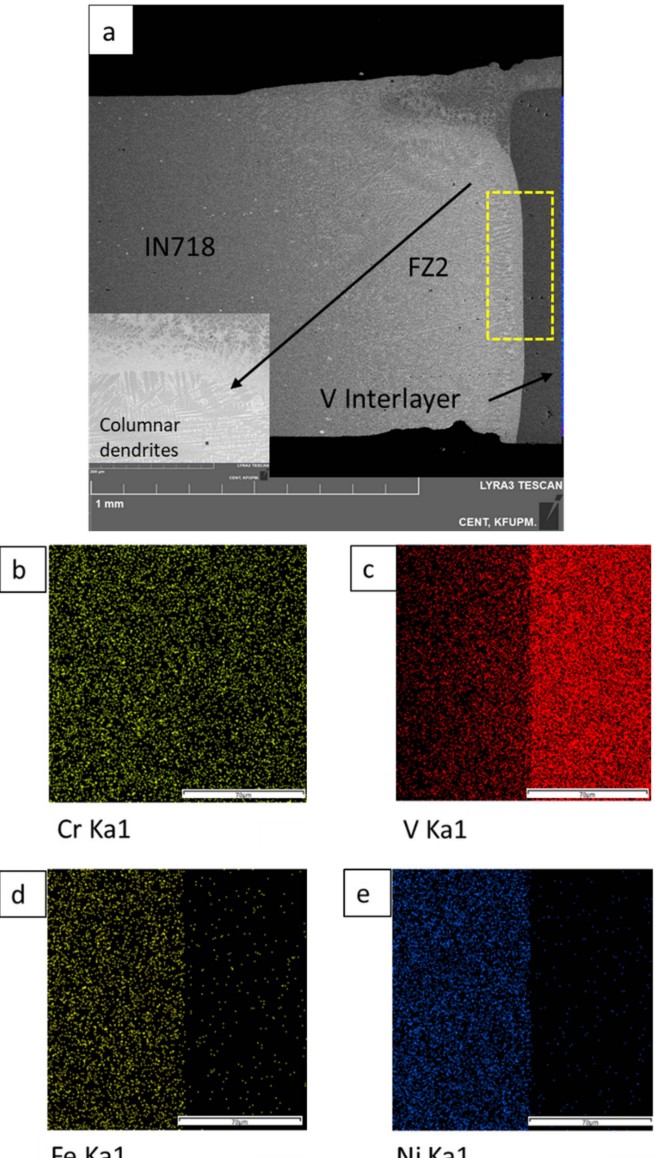

**Figure 3.** (**a**) Interfacial microstructures of V/IN718. Inset in (**a**) is a magnified image of the region indicated with the arrow. Elemental mapping of the region shown in (**a**) for (**b**) Cr, (**c**) V, (**d**) Fe, and (**e**) Ni.

The interfacial microstructure (FZ4 zone) and the elemental distribution of a selected area at the V/CpTi interface in Figure 4 shows the mutual diffusion of Ti and V. The diffusion of V in Ti resulted in the formation of a BCC (V,Ti) solid solution, due to the mutual solubility of both elements according to the EDS results, as shown in Table 2 and the Ti-V phase diagram. Similar results have been reported by Zoeram et al. [39].

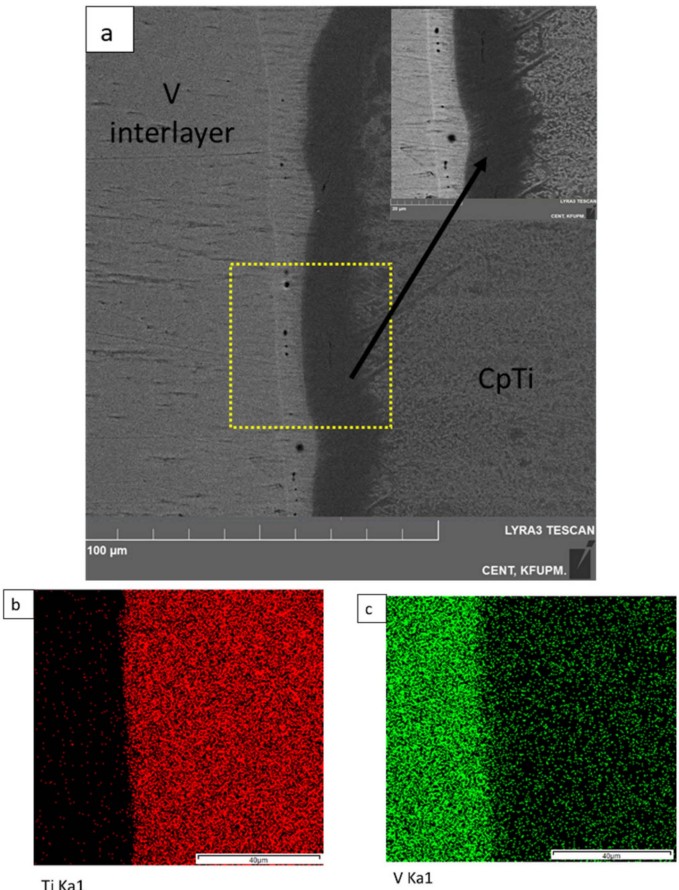

**Figure 4.** (**a**) Interfacial microstructures of V/CpTi interface. (**b**,**c**) Elemental mapping of highlighted region in (**a**).

The XRD spectra for the CpTi, IN718, and V joint is shown in Figure 5. Several IMCs, including NiV$_3$, Ti$_2$Ni, and TiNi, were detected in the melt zone and at the V/IN718 interface.

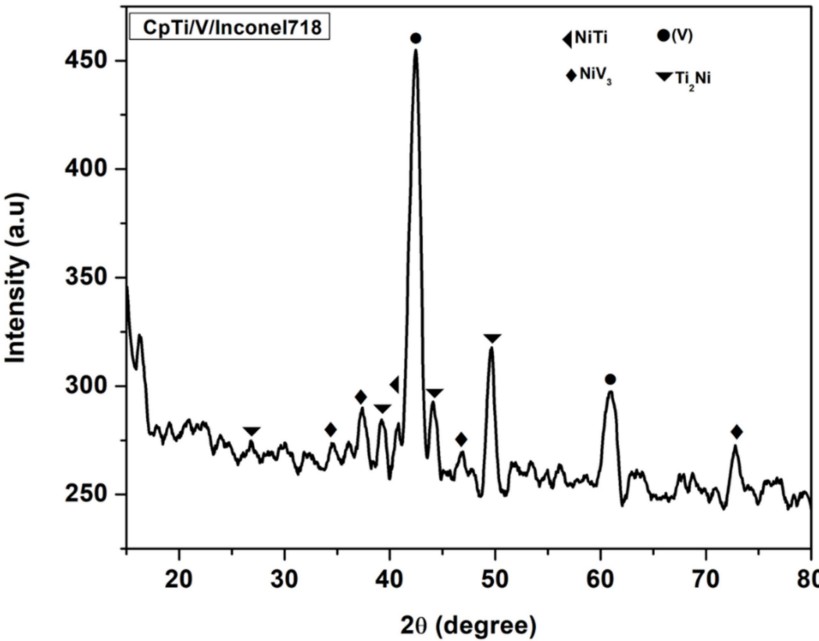

**Figure 5.** XRD spectra of the CpTi/V/IN718 joint.

*3.2. Nanoindentation*

3.2.1. CpTi and the Vanadium Interlayer Side

Nanoindentation tests were conducted on the base metal (BM) CpTi, fusion zones (FZ3 and FZ4), and the vanadium interlayer, as well as on the interfaces created during the welding of both alloys. Figure 6a shows the SEM image with indentation locations, while Figure 6b depicts the load vs. depth plots of CpTi with the vanadium interlayer. The deformation due to indentation consisted of the elastic-plastic loading and pure elastic unloading deformation. It can be seen that the V/CpTi interface (point B) has the lowest depth of penetration which has increased from 2100 nm to 2800 nm in the BM (CpTi). From this, it can be inferred that the V/CpTi interface has the least plastic deformation, although the indenter penetration depth in the BM (CpTi), V/FZ3 interface and the V interlayer is ~2800 nm. However, the unloading curve of the BM (CpTi) has shown the maximum elastic region compared to other areas of the welded joint. Therefore, the BM (CpTi) has the least elastic modulus.

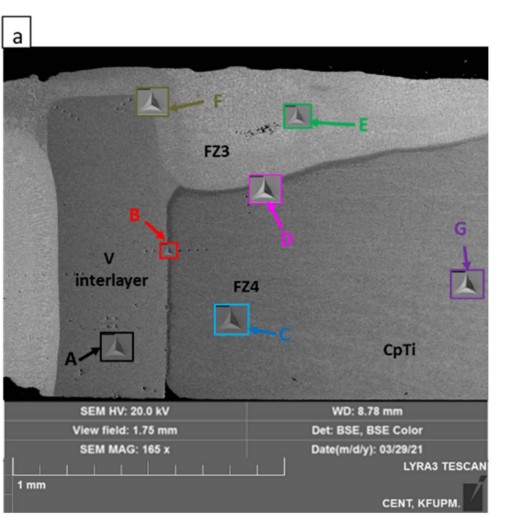 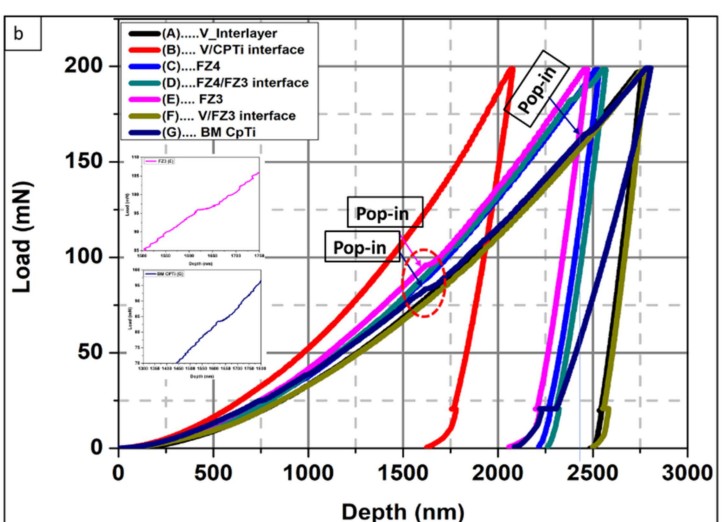

**Figure 6.** (**a**) SEM Image showing indentation on V/CpTi side. (**b**) Nano-indentation results of load vs. depth curves for CpTi and vanadium interlayer joint. Inset of (**b**) shows the pop-ins that appeared in the FZ3 and BM (CpTi) curves.

The occurrence of sudden discontinuities (pop-ins) in the loading curves of the FZ3 zone and BM (CpTi) are shown with arrows in Figure 6b. In the FZ3 region, a pop-in appeared at a penetration depth of about 1625 nm, whereas in the CpTi it appeared at a depth of approximately 1625 nm and 2400 nm. The formation of pop-ins can be attributed to the presence of grain boundaries in the indenter's stress field at a relatively large indentation load [40]. Plasticity initiation, and thereafter the elastic-to-plastic transition, often causes the first pop-in. The estimated critical shear stresses applied are on the order of the theoretical strength, indicating homogeneous dislocation nucleation. In future, further analysis can be carried out to characterize the pop-in behavior.

3.2.2. IN718 and the Vanadium Interlayer Side

Similarly, an SEM image showing the nanoindents and the load vs. depth curves of all possible locations in the IN718 and vanadium joint region are presented in Figure 7a,b, respectively. Significant differences in maximum penetration depth can be observed across all the regions. This shows that the indentation response is highly distinguishable from the details of the metallurgical conditions resulting from the welding process. The V/IN718 interface showed indenter penetration at a depth of approximately 1250 nm, which increased to ~1600 nm, ~1900 nm, ~2400 nm and ~2750 nm in the FZ1 (common melt zone), FZ2, BM, and V interlayer, respectively. Therefore, the V/IN718 interface is brittle and has the least plastic deformation. In addition, the V/IN718 interface showed

an indenter penetration 40% lower than the V/CpTi interface, indicating that the joint is more brittle.

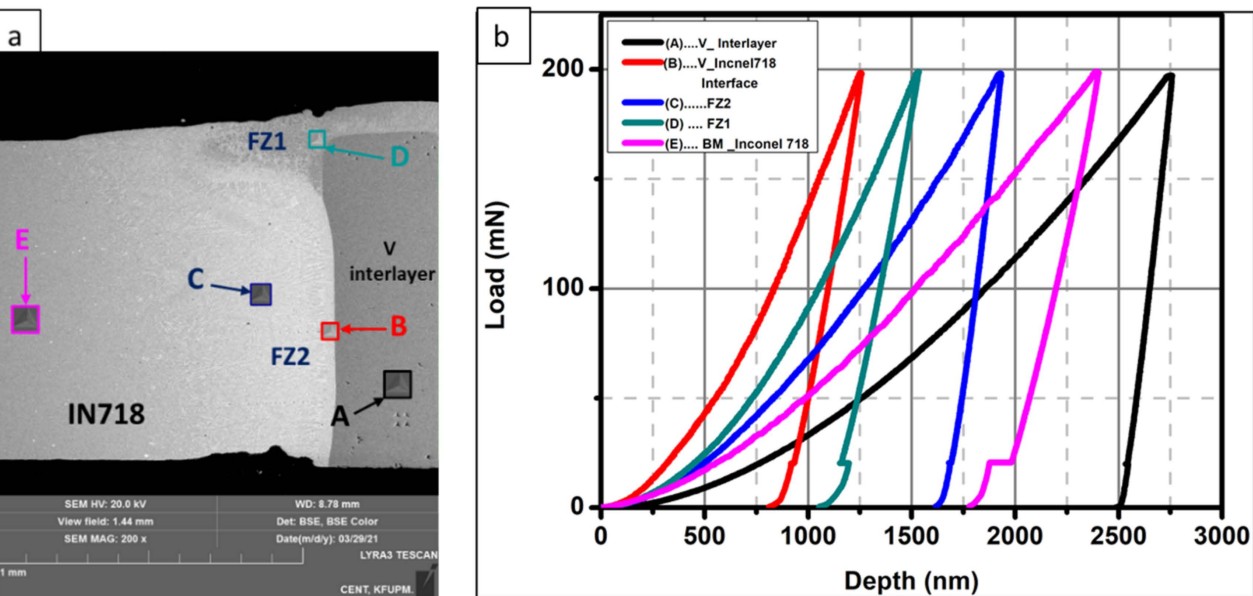

**Figure 7.** (**a**) SEM images showing nano indents on the V/IN718 side. (**b**) Nano-indentation results of load vs. depth curves for (N718) and the vanadium interlayer joint.

### 3.3. Nanohardness Line Scan

A nanohardness line scan is a useful technique for studying transitions in characteristics across the fusion boundary and the interfacial properties in dissimilar alloy weldments. Indentation line scans have been carried out at different interfacial regions marked at locations A, B, C, D, and E, as shown in Figure 8. The distance between the indents was kept to 40 nm at a load of 200 mN. Location (A) in Figure 9a shows the transition zone between the V interlayer and the IN718 alloy, along with the indentation sequence. The corresponding nanohardness line scans and elastic moduli are presented in Figure 9b.

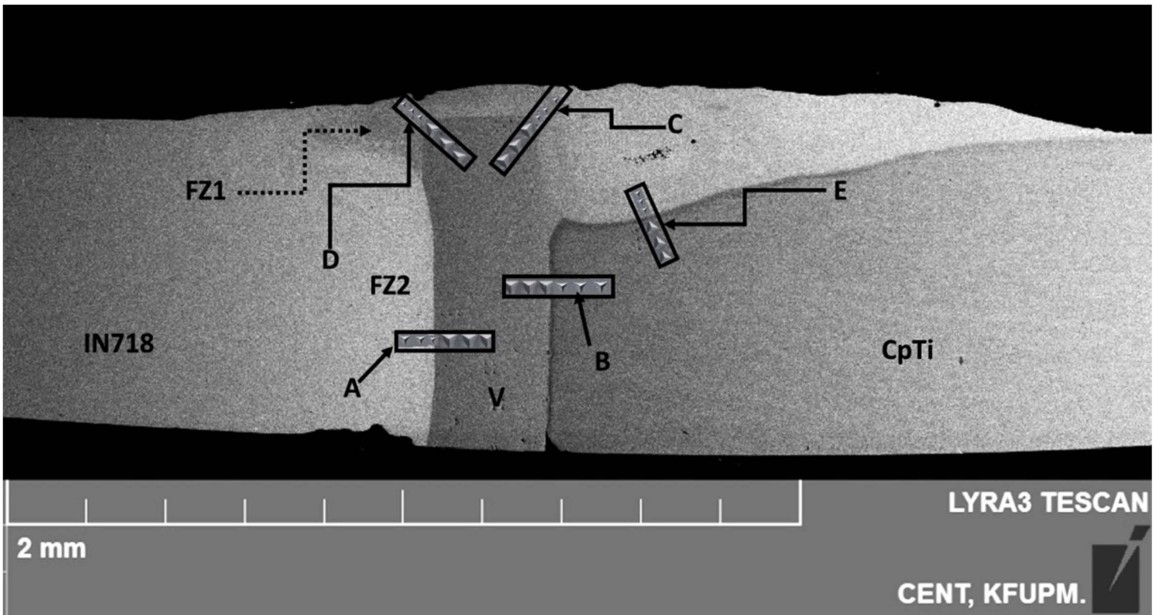

**Figure 8.** Locations selected for nanohardness line scans across the CpTi/V/IN718 joint.

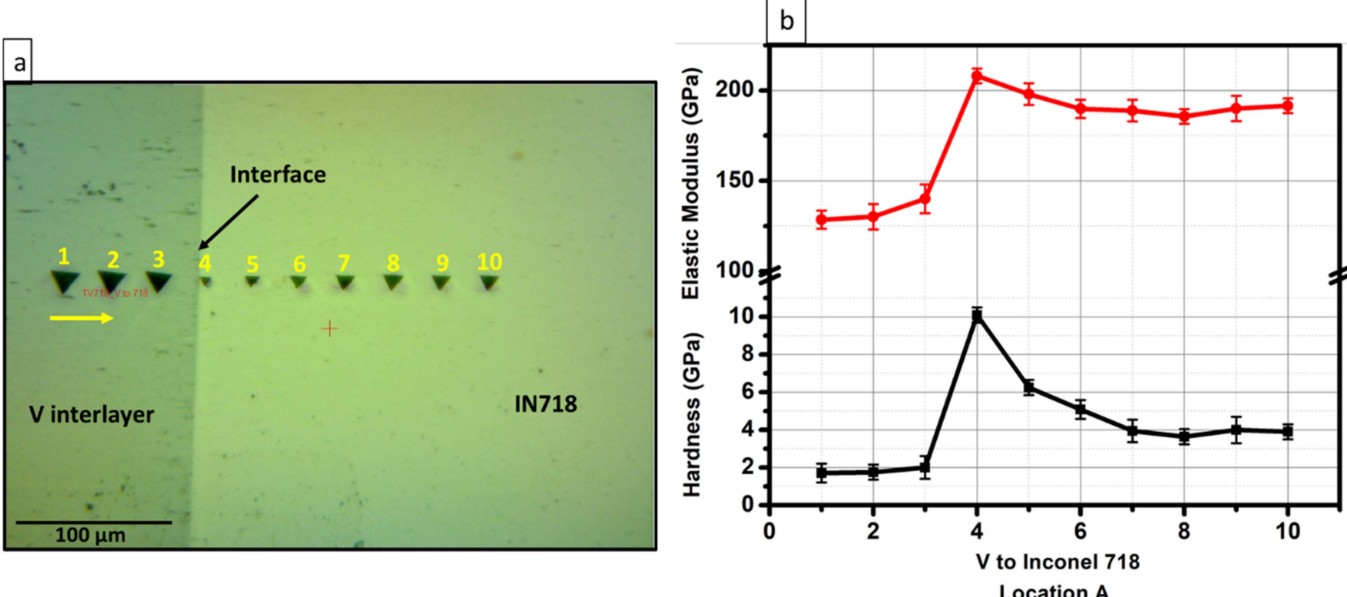

**Figure 9.** (**a**) Nanohardness line scan at the V/IN718 Interface (location (A) in Figure 8)). (**b**) Corresponding hardness and elastic modulus profiles along the line scan.

A hardness peak as high as approximately 10.1 GPa can be clearly seen at the interface. This gradually decreases to ~6.3, ~5.1, and ~4 GPa towards the IN718 side, whereas ~1.8 GPa was observed in the V interlayer. Furthermore, the same hardness values were observed throughout the whole interface between the V interlayer and IN718. The sizes of the indentations also match the variation in hardness (lower indentation size means higher hardness and vice versa). The reason behind the variation in hardness could be due to the formation of IMCs. The presence of $NiV_3$ IMCs at the interface has been confirmed by the EDS and XRD results. The similar hardness values for Ni- and V-based IMCs are reported in [38]. In addition, it can be seen in Figure 9b that the elastic modulus of ~130 GPa in the V interlayer has increased to as high as ~210 GPa at the interface, then decreased gradually to a minimum value of ~190 GPa. The reason behind the variation in hardness could be attributed to the existence of IMCs. The existence of $NiV_3$ IMCs at the interface have been confirmed by the EDS and XRD results. The same hardness trends of Ni- and V-based IMCs are also reported in [38].

Location (B) was the transition zone between the V interlayer and CpTi as shown in Figure 10a. A maximum hardness of ~3.3 GPa was observed at the interface between the two alloys, which remained constant at ~2 GPa towards the CpTi region (Figure 10b). The increase in hardness at the interface can be attributed to the formation of a brittle BCC (Ti, V) solid solution through mutual diffusion of Ti and V. The creation of a Ti, V solid solution is also indicated by the EDS results.

The elastic modulus in the V interlayer was as high as ~131.8 GPa, then reduced to a minimum value of ~107.6 GPa, followed by a gradual decrease to ~112 GPa in the CpTi region. It can be seen from the line profile that hardness and elastic modulus could not follow the same trend. This can be attributed to the intrinsic nature of the elastic modulus of any material, which is related to the atomic bonding, whereas hardness is an extrinsic materials response related to yield strength [41]. The hardness at the V/IN718 interface is approximately 67.7% greater than that at the V/CpTi interface.

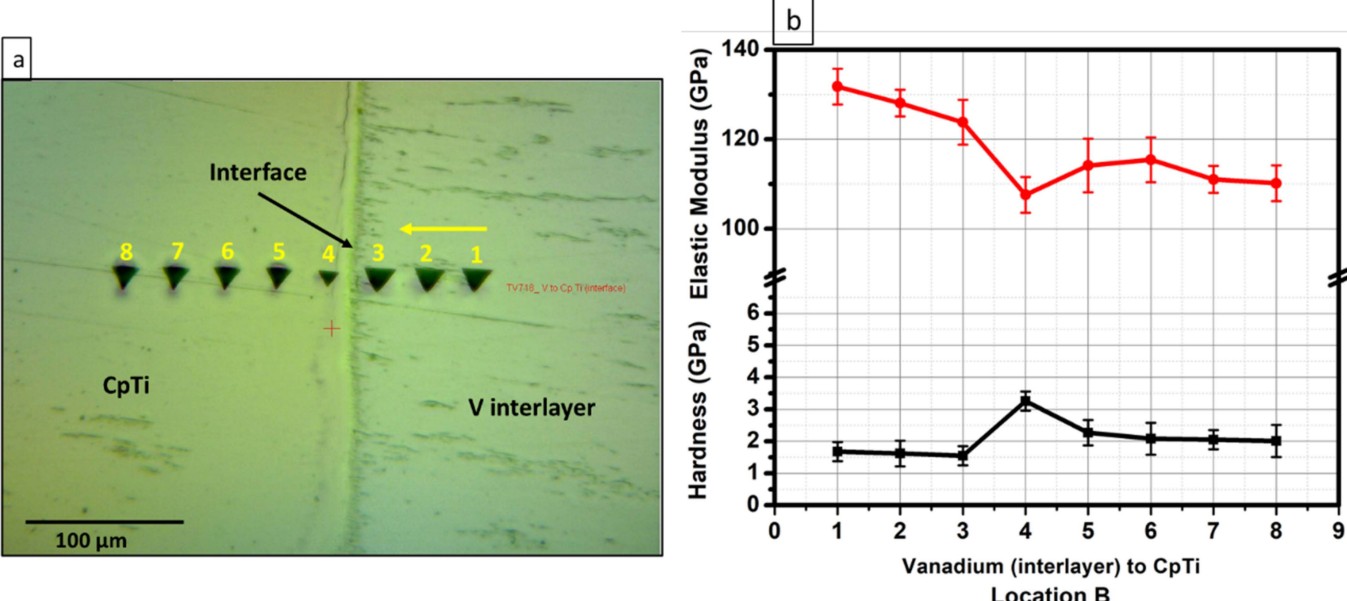

**Figure 10.** (**a**) Nanohardness line scan at V/CpTi Interface (location (B) in Figure 8). (**b**) Corresponding hardness and elastic modulus profiles along the line scan.

Location (C), the zone observed between the V interlayer and the common melt zone (FZ3), can be seen in Figure 11. This profile presents two distinct regions instead of an interface with very high hardness values. It can be clearly seen that the hardness varied from ~1.7 GPa in the V interlayer to an average of ~7.5 GPa in the FZ3 region. The existence of a high hardness region can be attributed to the formation of $Ti_2Ni$ brittle IMCs. The formation of the same phase has also been confirmed by the XRD results. However, the elastic moduli of ~122 GPa and ~125 GPa were observed in the V interlayer and FZ3 region, respectively.

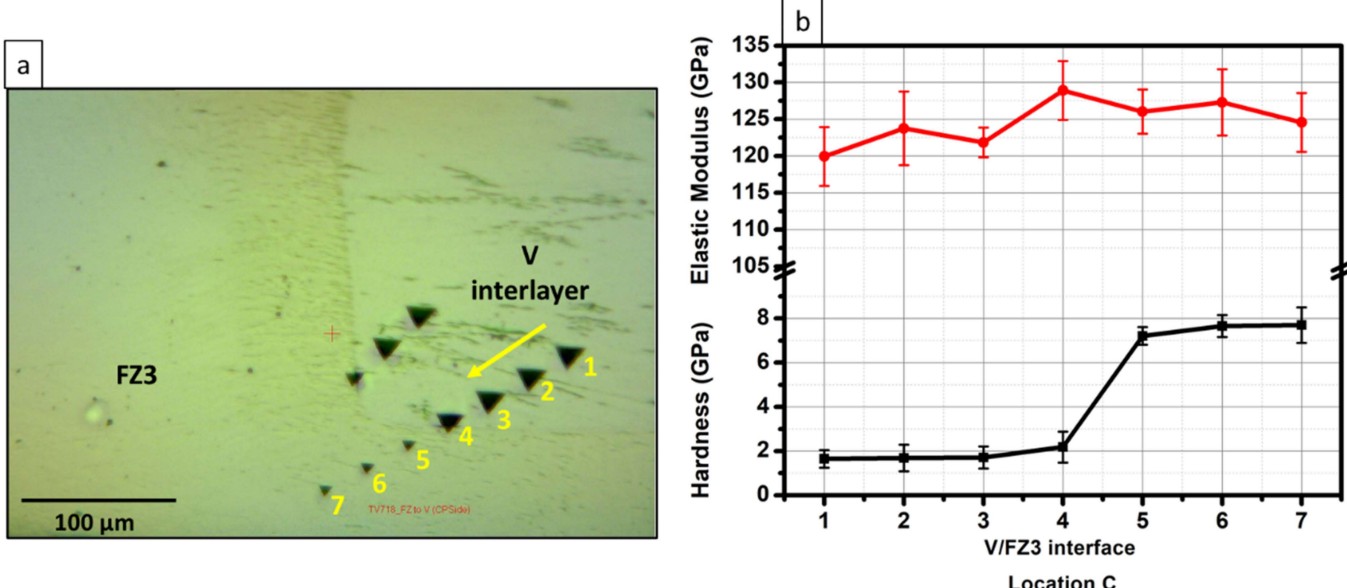

**Figure 11.** (**a**) Nanohardness line scan at V/FZ3 Interface (location (C) in Figure 8). (**b**) Corresponding hardness and elastic modulus profiles along the line scan.

Location (D) is the region between the V interlayer and the fusion zone (FZ1), as shown in Figure 12. The hardness in the vanadium interlayer varied from ~1.6–1.8 GPa,

whereas ~6.5–8.1 GPa was observed in the FZ1. The high hardness in the fusion zone indicates the presence of Ti$_2$Ni, NiV$_3$, and NiTi IMCs. However, the elastic modulus found in both regions ranges from ~130–134 GPa and ~132–136 GPa in the V interlayer and FZ1, respectively. The nanohardness in the FZ1 zone on the IN718 side is approximately 8% higher than that in the FZ3 zone on the CpTi side.

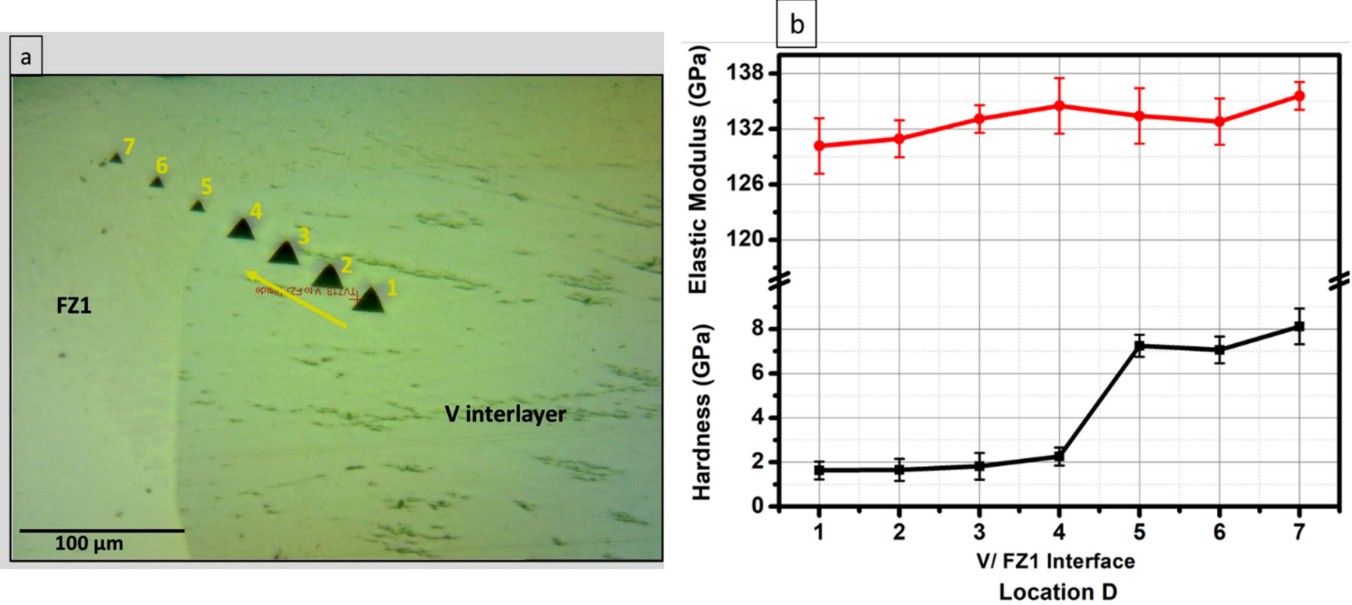

**Figure 12.** (**a**) Nanohardness line scan at V/FZ1 Interface (location (D) in Figure 8). (**b**) Corresponding hardness and elastic modulus profiles along the line scan.

There is another distinct region (Figure 13), marked as location (E), which is a transition between the fusion zones FZ4 and FZ3 with no vanadium interlayer. Nanohardness in the FZ4 region was ~2.1 GPa with an elastic modulus of ~114 GPa, whereas the FZ3 region was observed to have a high hardness of 6.4 GPa and an elastic modulus of 122 GPa.

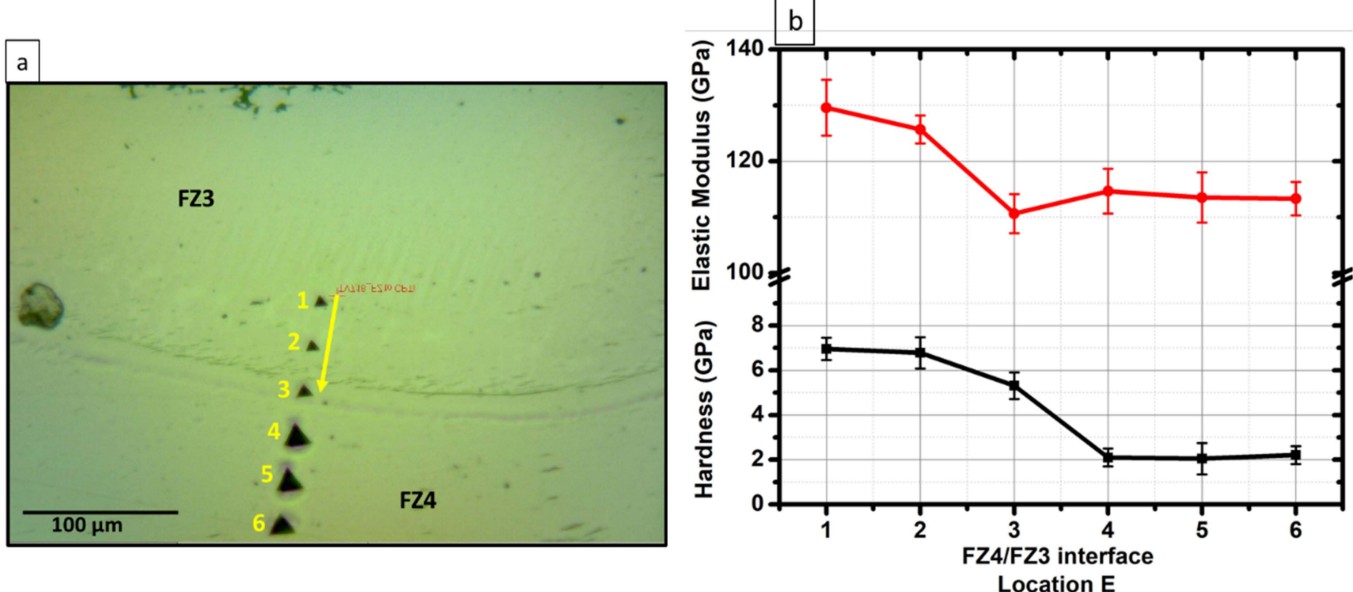

**Figure 13.** (**a**) Nanohardness line scan at FZ3/FZ4 Interface (location (E) in Figure 8). (**b**) CorreScheme 718. interface has a relatively higher nanohardness, approximately 30% greater than FZ2 (which has the highest hardness after the V/IN718 interface) and 70% greater than the IN718 base metal.

The different weld regions and interfaces showed that the V/IN718 interface has a relatively higher nanohardness, approximately 30% greater than FZ2 (which has the highest hardness after the V/IN718 interface) and 70% greater than the IN718 base metal.

### 3.4. Mechanical Properties Mapping

Figure 14a shows a nanohardness distribution map at the V/IN718 joint. The analysis grid contains $45 \times 35 = 1575$ indents in $140~\mu m \times 70~\mu m$. The neighboring indents distance is $3~\mu m$ and $2~\mu m$ in the x-axis and y-axis directions, respectively. To make a comparison, the same indentation grid was applied at the V/CpTi side. The nanohardness increased abruptly to the highest value of approximately 17 GPa on the IN718 side/FZ2 region (the area shown in red), whereas the lowest hardness of approximately 2 GPa (the area shown in violet) can be seen in the V interlayer. This increase in hardness can probably be associated with the presence of IMCs ($Ti_2Ni$, $NiV_3$, and TiN). The nanohardness was found to be nonuniform in the IN718 side (the red area). Owing to the solidification segregation during cooling, the nanohardness in the columnar dendritic zone was inhomogeneous. The width of this highest hardness region was approximately $60~\mu m$, which can be allied with the diffusion of V in the IN718 side.

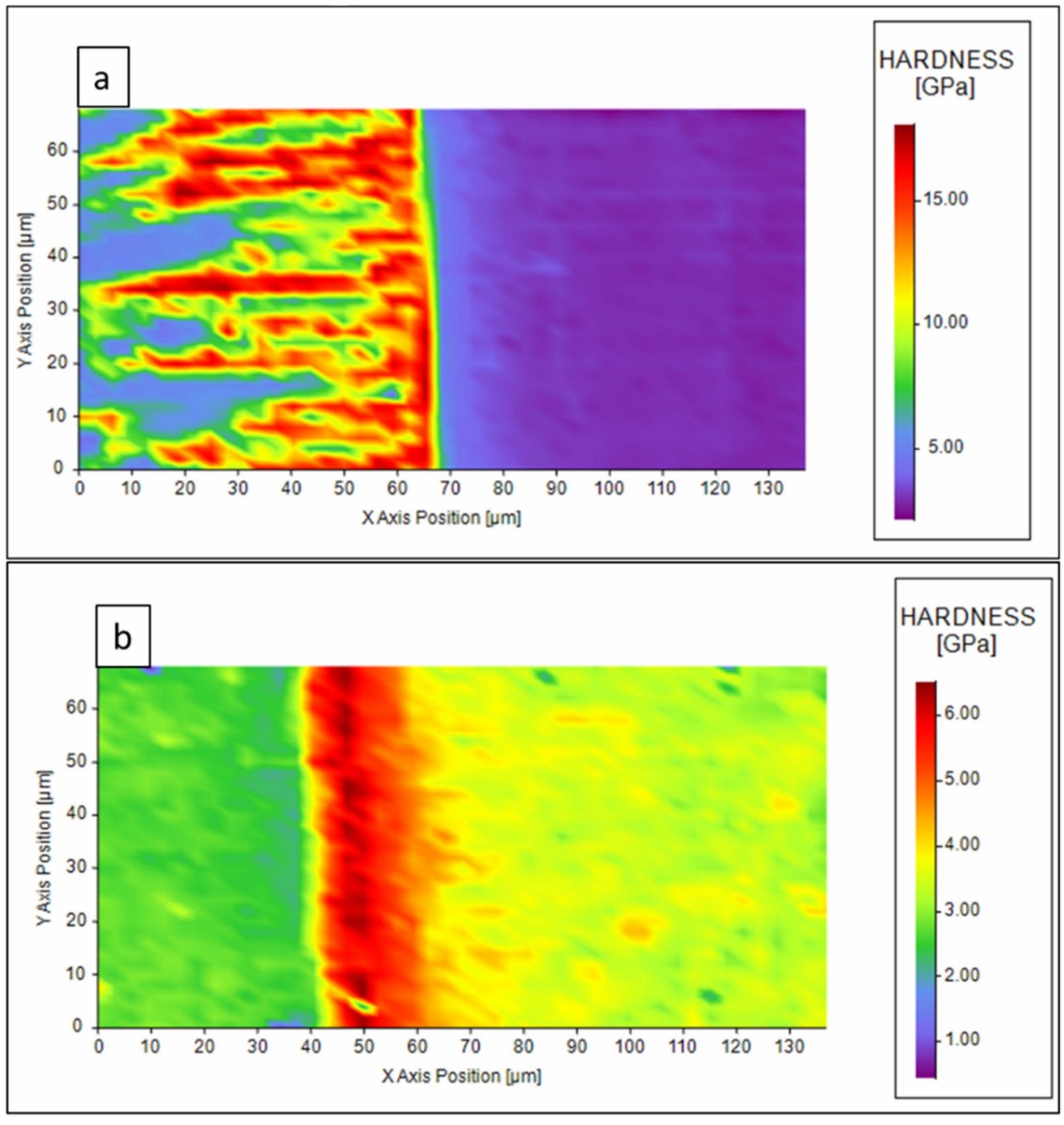

**Figure 14.** Nanohardness maps. (**a**) The V/IN718 interface. (**b**) The V/CpTi interface.

On the other hand, the V/CpTi interface was observed to have a nanohardness ranging from ~4.5 GPa to 6.2 GPa (area in red) with a width of approximately 17 μm, as shown in Figure 14b. In addition, this region shows a homogenous hardness distribution at the interface which can be associated with the mutual solubility of Ti and V. Moreover, the increase in hardness at the interface can be attributed to the formation of the (Ti, V) solid solution. The nanohardness values in the V interlayer and on the CpTi side were found to be ~3 GPa and ~4 GPa, respectively.

### 3.5. Nano Creep

### 3.5.1. V/CpTi Side

Typical load–displacement (*P–h*) curves of V/CpTi interface, FZ4, FZ3, the V/FZ3 interface, and BM in CpTi and V interlayer side are illustrated in Figure 15a. It can be seen that all the P–h curves show a similar trend in the loading regime and that indenter displacement increases from approximately 1200 nm in the FZ3 region to 2800 nm in the V interlayer region. The corresponding creep displacement–time (*h–t*) curves taken from the holding regime of different weld regions in the nanoindentation tests of the V interlayer and on the CpTi side are shown in Figure 15b. In the holding regime, the P–h curves in different weld regions exhibited significant load plateaus, i.e., the creep displacement reaches a maximum value of approximately 280 nm in the V/FZ3 interface under the load of 200 mN and this value is ~100 nm in FZ3 and at the FZ4/FZ3 interfacial region. The widening of load plateaus in the holding regime implies a relatively greater amount of creep deformation.

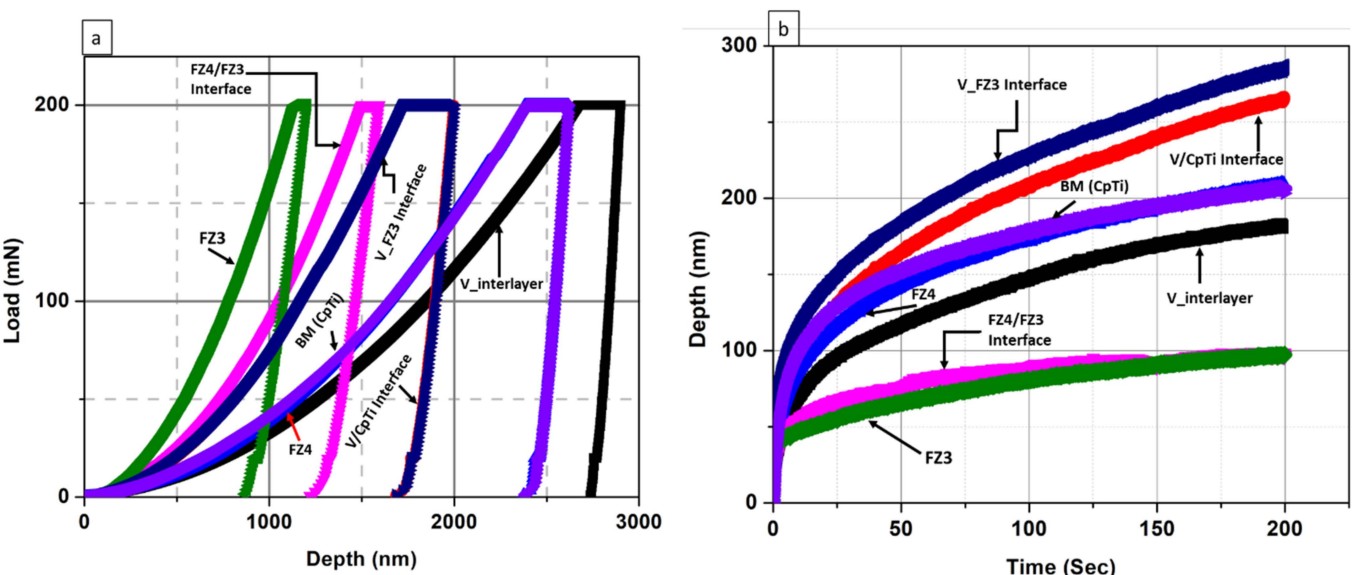

**Figure 15.** (**a**) The load–displacement (P–h) curves of different weld regions of V and CpTi obtained under 200 mN at 200 sec holding time. (**b**) The displacement–time (h–t) curves obtained in the holding regimes of different regions of V and the CpTi side.

It is visible from the curves that the FZ3 and FZ4 regions show creep displacement of ~100 nm and 220 nm, which are ~56.5% and ~4.3% less than the BM (CpTi). Similarly, creep displacements at the interfaces (V/CpTi and V/FZ3) are ~47.2% and ~55.5% higher than that at the V interlayer, respectively, whereas the FZ4/FZ3 interface shows ~44.4% less creep displacement than the V interlayer.

The variation in creep depth can be attributed to the diffusion of CpTi and V, with the formation of IMCs in the common melt zone of all three alloys on the CpTi side.

In comparison to the BM and V interlayer, on the CpTi side, the FZ3 mainly comprises $Ti_2Ni$, TiNi, and $Ni_3V$, which are of high hardness ($Ni_3V$ > 10 GPa) and consequently show resistance to indenter penetration, resulting in lower creep rates. Similarly, the presence of

Ti$_2$Ni at the FZ4/FZ3 interface also shows less penetration. However, the formation of the (Ti, V) solid solution at the V/CpTi interface reduces the hardness to 3.5 GPa, resulting in a higher creep deformation. It is worth mentioning that the creep deformations of the BM and V interlayer are relatively lower than that of the V/FZ3 interface and other regions. This deformation behavior can be attributed to the relatively fine grain size of both alloys. Lou et al. [42] processed CPTi by equal channel angular pressing (ECAP) and reported good creep properties of ultra-fine-grained CPTi at room temperature.

It can also be seen that all the curves have a similar increasing tendency, i.e., the indentation depth increases rapidly with increasing holding time at the initial stage (transient creep regime) for each zone, and then increases slowly or almost linearly with an increasing holding time at the later stage (steady-stage creep regime). However, the tertiary creep stage does not appear in the indentation creep, unlike the conventional creep. This happens because the hardness test is basically a compression test and the specimen does not fracture, making it impossible to quantify a third stage of the curve [43]. The fitted curves for the experimental data in the V/FZ3 and BM regions of the CpTi using Equation (4) are presented in Figure 16. In the curves, the black square symbols indicate the experimental data, while the red line represents the fitted curve. It can be seen that all the regions showed good fitting for the creep depth vs. time curves.

ln (creep strain rate)-ln (stress) are presented in Figure 17a for indentation tests in different regions of the V interlayer and the CpTi side. The slope of the linear function is the creep stress exponent in the linear fitting. Creep stress exponent values indicate a material's creep behavior in terms of creep resistance and the active creep mechanism. The creep stress exponent of 1 indicates diffusion creep, the creep stress exponent of 2 indicates the activation of the grain boundary sliding mechanism, and the creep stress exponents of 3 and 4–5 indicate the activation of the viscous motion of dislocations and the dislocation climb mechanism, respectively [44,45]. In addition, the variation in creep stress exponent values implies a change in creep resistance across regions. A higher creep stress exponent value indicates a greater creep resistance [34].

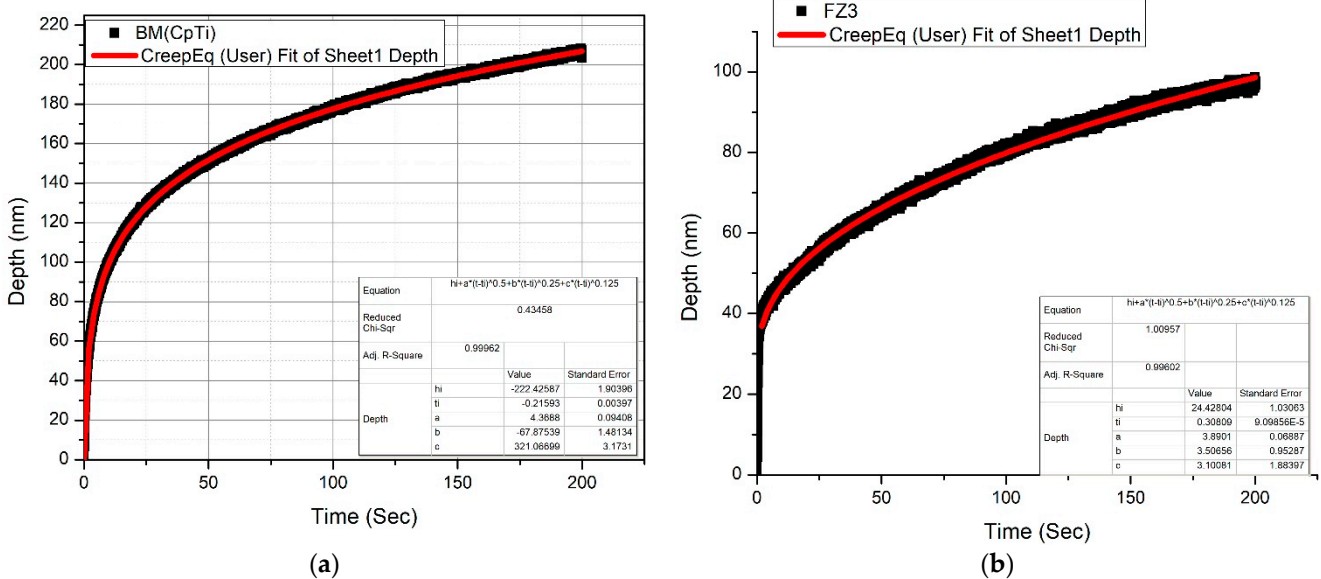

**Figure 16.** *Cont.*

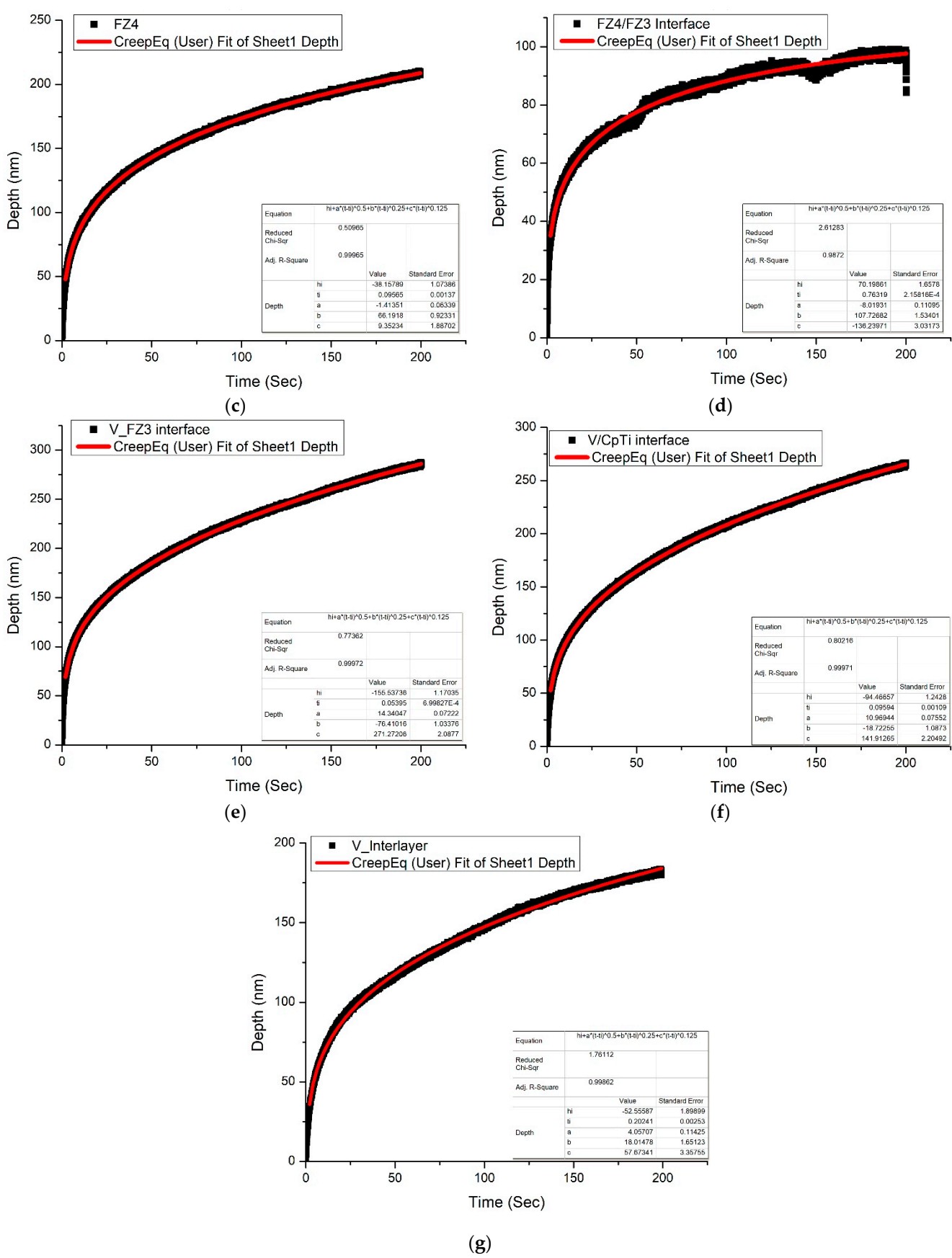

**Figure 16.** Experimental data and fitted curves for: (**a**) BM of the CpTi side; (**b**) FZ3; (**c**) FZ4; (**d**) the FZ4/FZ3 interface; (**e**) the V/FZ3 interface; (**f**) the V/CpTi interface; and (**g**) the V interlayer.

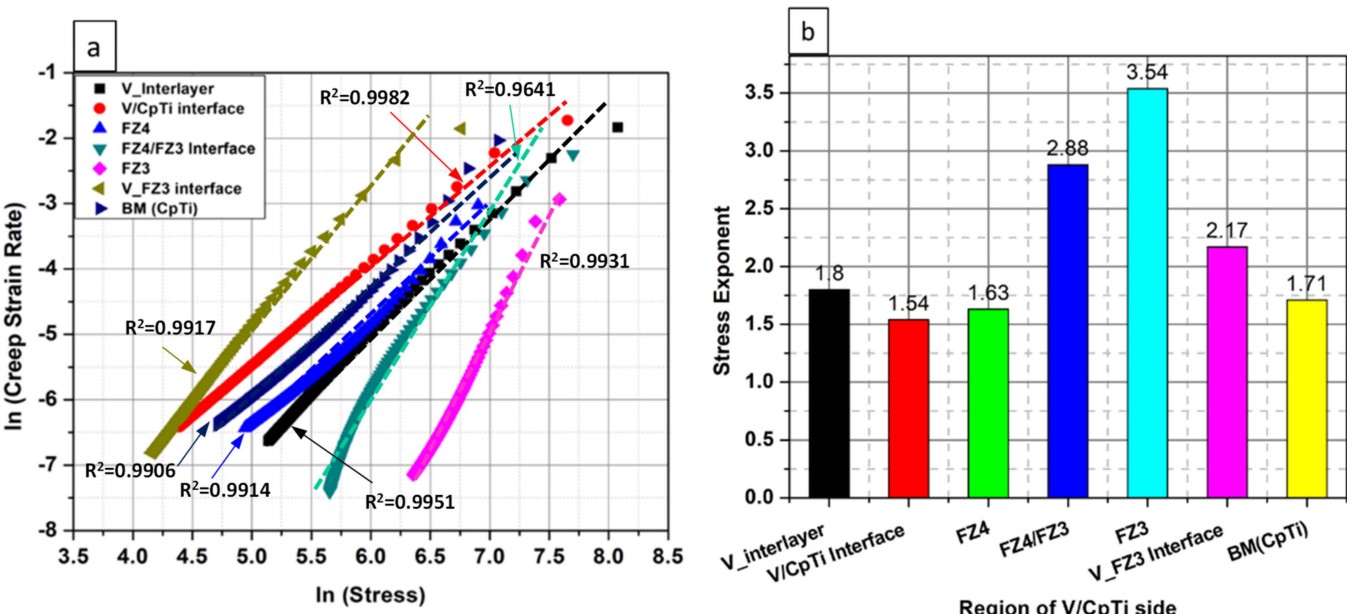

**Figure 17.** (**a**) ln (creep strain rate)-ln(stress) plots of different regions of V and CpTi side. (**b**) Creep stress exponent of different regions in V and the CpTi side.

Through calculation, and with the use of Equation (7), the stress exponent in the V interlayer, the V/CpTi interface, FZ4 and BM (CpTi) was found to be less than 2, as shown in Figure 17b. As a result, the active creep mechanism in these regions is a diffusional one. This could be due to the diffusion of vacancies, that caused plastic deformation of the material over time. The creep behavior of the Inconel 617 top layer was investigated by Salari et al. [46] and the diffusion was found to be the active creep process with a creep stress exponent of less than 2 at ambient temperature. However, the creep mechanism at the interfaces of V/FZ3 and FZ4/FZ3, with higher creep stress exponents (2.17 and 2.88), might be caused by the grain boundary sliding. Similar plastic deformation by the grain boundary sliding mechanism at the interfaces in nanocyrstalline FCC metals is reported [47]. Moreover, a stress exponent in the FZ3 region greater than 3 represents a dislocation climb mechanism showing the highest creep resistance over all the regions.

3.5.2. V/IN718 Side

In the V/IN718 side, typical load–displacement (P–h) curves of different regions viz, the V/IN718 interface, FZ1, FZ2, the V interlayer and BM/IN718 are shown in Figure 18a. It can be seen from all the P–h curves that indenter penetration increases to approximately 1200 nm in the V/IN718 interface and to 2900 nm in the V interlayer. Figure 18b depicts the corresponding creep displacement–time (h–t) curves obtained from the holding regime of 200 s in different weld locations of the joint at the V interlayer and the IN718 side. Moreover, in the holding regime, all the curves displayed considerable load plateaus in different regions, i.e., the creep displacement reaches a maximum value of approximately 220 nm in the V interlayer, which is ~50 nm in the FZ1 region.

It can be seen from the curves that the creep displacements of FZ1 and FZ2 were ~50 nm and 105 nm, respectively, which was 54.5% and 4.5% less than the BM (IN718). Similarly, creep displacement at the V/IN718 interface was 31.7% less than the V interlayer and 27.2% higher than the BM (IN718).

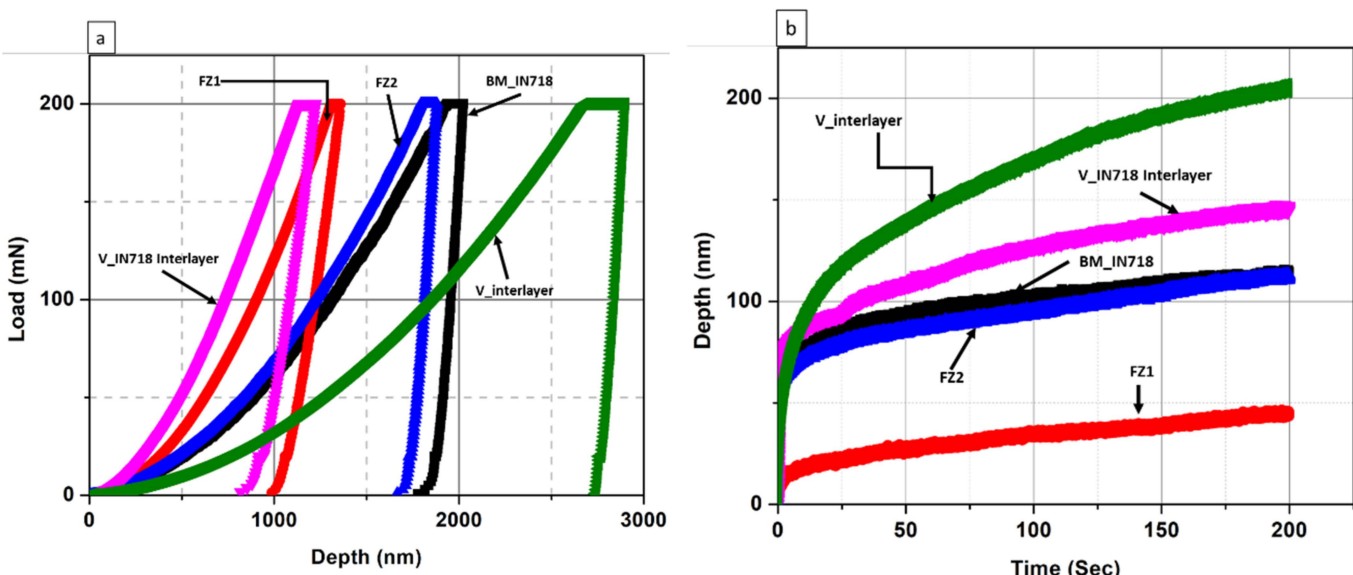

**Figure 18.** (**a**) The load–displacement (P–h) curves of different regions of V and IN718 obtained under 200 mN at 200 sec holding time. (**b**) The displacement–time (h–t) curves obtained in the holding regimes of different regions of V and the IN718 side.

The existence of IMCs in distinct zones of the IN718 side can explain the variation in creep depth. In comparison to the BM and V interlayer in the IN718 side, the FZ1 region, which is primarily composed of $Ti_2Ni$, TiNi, and TiN, and has a high hardness (TiN > 13 GPa), exhibited resistance to indenter penetration, resulting in decreased creep deformation rates. Moreover, the high creep strength can be attributed to a long-range order of the intermetallics, which slows down the diffusivity rate. In addition, a decrease in the diffusion rate would result in a decrease in the creep rate in alloys where the dislocation climb is a rate-controlling parameter [48].

However, it can be clearly seen that the V/IN718 interface, being harder (where $NiV_3$ IMCs having hardness more than 10 Gpa), has shown a higher penetration depth (less resistance to indentation). This could have happened because the indentation took place in the spaces between the precipitates or NiTi IMCs, resulting in higher creep rates [49]. The fitted curves for the experimental data in the FZ1 and BM regions of IN718 are presented in Figure 19. As can be seen from the creep depth vs. time graphs, all zones fitted well.

The ln (creep strain rate)-ln (stress) curves of different regions of the V interlayer and the IN718 side are shown in Figure 20a. Bar charts shown in Figure 20b present the values of the stress exponents. The stress exponents determined from indentation curves ranged from 2.52 to 4.15 and indicate the active creep mechanism in FZ1, comprised of IMCs, through dislocation climb. The magnitudes of the dislocation densities produced in the metals and IMCs are the same. The different crystal structure of the IMCs created during the welding process would require different applied loads to initiate the occurrence of sliding and dislocation [50,51]. As a result, the IMC phases, with their various crystal structures, will have different micromechanical characteristics and exhibit different creep behavior. Moreover, changes in the stress exponent values could be caused by microstructural changes in the weld zones, such as changes in lattice characteristics, the solid solution, the size and distribution of strengthening phases, and the intermetallic phases [52].

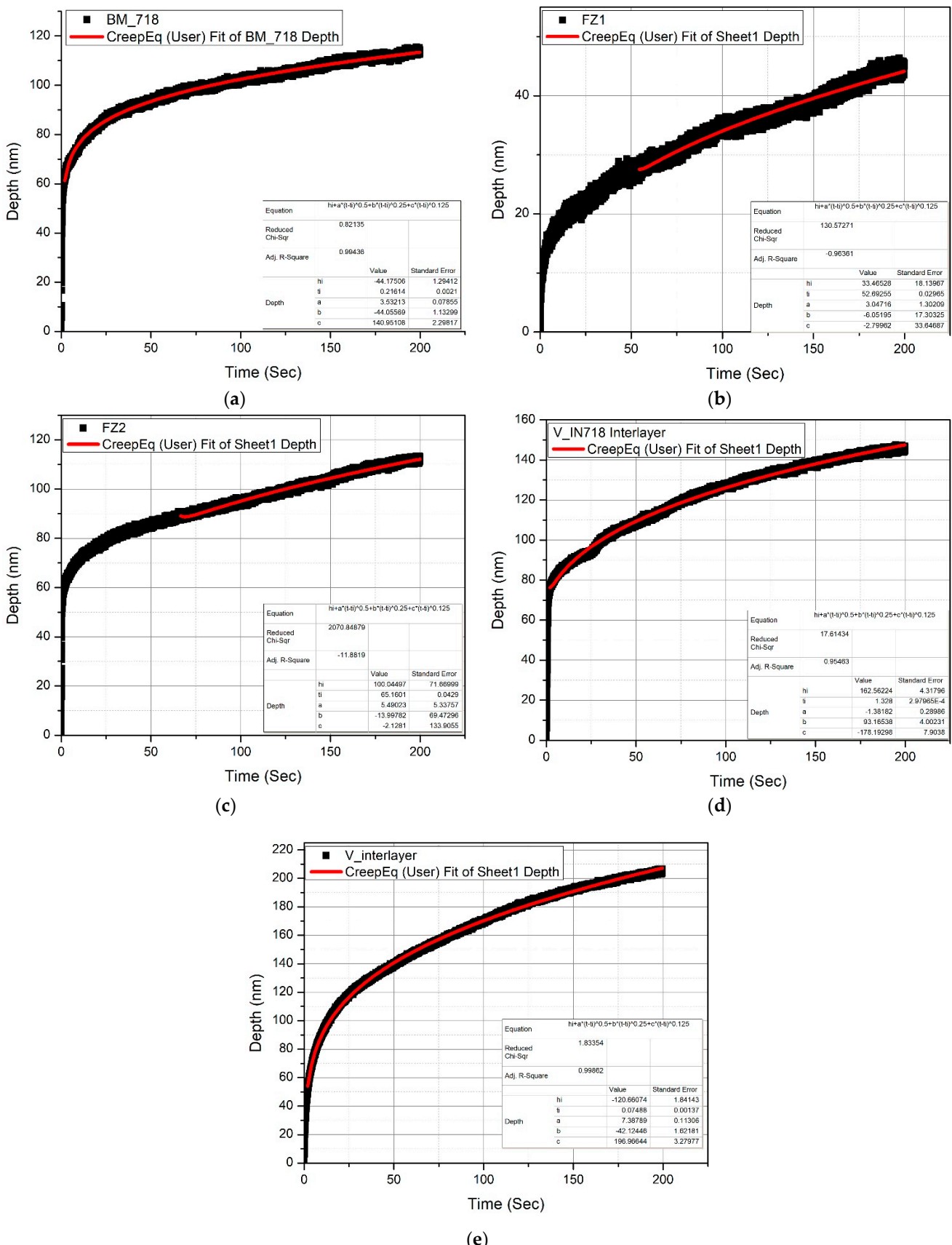

**Figure 19.** Experimental data and fitted depth–time curves for: (**a**) BM/IN718; (**b**) FZ1; (**c**) FZ2; (**d**) V/INC 718; and (**e**) the V interlayer.

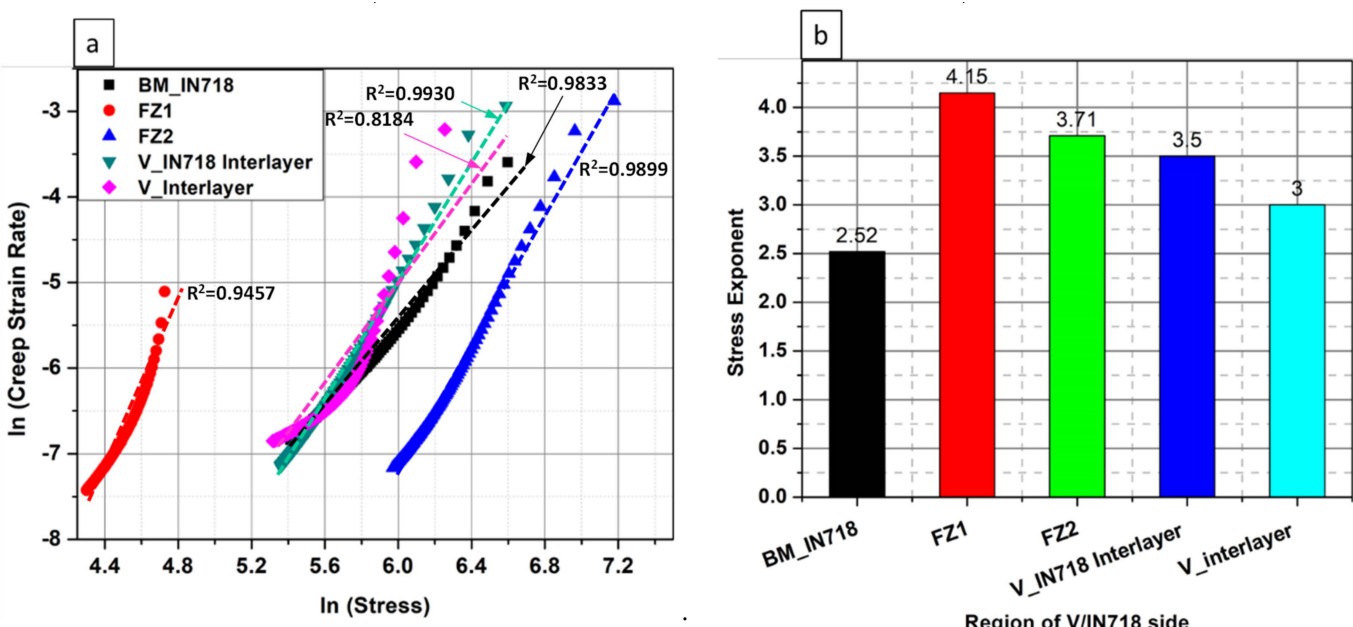

**Figure 20.** (**a**) The ln (creep strain rate)-ln (creep stress) curves obtained under the peak load in different regions of IN718. (**b**) Creep stress exponent of different regions in V and the IN718 side.

### 3.5.3. Creep Discussions

Creep at low temperatures can be explained in terms of time-dependent plasticity occurring at T < 0.3 Tm and stresses that are frequently less than the macroscopic yield stress (offset yield point at 0.2%). At lower temperatures, several materials exhibit significant plasticity. In addition, low temperature creep includes the validity of the mechanism that involved the intersection of dislocations.

At high temperatures, creep in metals is a time-dependent inelastic deformation that occurs when the metal's temperature increases to more than 0.3 times its melting temperature. Furthermore, in high-temperature creep, plastic flow occurs as a result of three unique types of creep mechanisms: (i) intragranular dislocation creep, (ii) grain boundary sliding, and (iii) diffusion creep. Creep rate at high temperature depends on testing temperature via the diffusion coefficient and the magnitude of the applied stress [31].

The findings are regarded to be significant in respect of the localized creep behavior of dissimilar welds at relatively low temperatures and the indication given for radical revision to the recommendations for designers when it comes to selecting material combinations for welded structures conducive to the maximization of component lifetimes based on the relevant operating loads and temperature. This research has the potential to show how to reduce the weight of thin-walled structures by opening up a wide range of possibilities for welding different metals and alloys together with an interlayer.

### 4. Conclusions

In this paper, nanohardness profiles across different weld regions and room temperatures and nanoindentation creep of TIG welded to a CpTi/V/IN718 joint have been investigated.

Due to the formation of IMCs ($NiV_3$ and $Ti_2Ni$), the V/IN718 interface and FZ2 zone showed the highest nanohardness in comparison to the other regions of the weldment, which is approximately 30% greater than FZ2 and about 70% higher than the IN718 base metal.

At the vanadium and CpTi side, the FZ3 and FZ4 regions showed creep displacements of approximately 56.5% and 4.3% less than that at the BM (CpTi), respectively. Similarly, the creep displacement at the interfaces (V/CpTi and V/FZ3) were about 47.2% and 55.5% higher than that at the V interlayer, respectively.

At the vanadium and IN718 side, the creep displacements of FZ1 and FZ2 were approximately 54.5% and 4.5% less than the BM (IN718), while the creep displacement at the V/IN718 interface was about 31.7% less than the V interlayer and about 27.2% higher than the BM (IN718).

The stress exponents determined from the indentation curves on the V/CpTi side ranged from 1.54 in the V/CpTi interface to 3.54 in the FZ3 region. The creep mechanism at room temperature was diffusional creep, and dislocation glide dominated the creep deformation.

The creep stress exponent on the V/IN718 was relatively higher (between 2.52 and 4.15) than the V/CpTi side due to the presence of IMCs with high hardness, which decreased the diffusivity rate because of the long-range order of the intermetallics.

**Author Contributions:** Conceptualization, T.S., F.N.K. and M.J.; methodology, T.S., F.N.K. and M.J.; validation, T.S., F.N.K., M.J. and J.H.; formal analysis, T.S., F.N.K., M.J. and J.H.; investigation, T.S., F.N.K. and M.J.; writing—original draft preparation, T.S., F.N.K., M.J. and J.H.; writing—review and editing, T.S., F.N.K., M.J. and J.H.; supervision, F.N.K., M.J. and J.H.; visualization: T.S., F.N.K., M.J. and J.H. All authors have read and agreed to the published version of the manuscript.

**Funding:** This research received no external funding.

**Data Availability Statement:** The data presented in this study are available within the article.

**Acknowledgments:** The authors acknowledge the support from the Bangladesh Jute Research Institute and the Bangladesh University of Engineering and Technology, Dhaka, Bangladesh.

**Conflicts of Interest:** The authors declare no conflict of interest.

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
