# Peer review of "Investigating Nanoindentation Creep Behavior of Pulsed-TIG Welded Inconel 718 and Commercially Pure Titanium Using a Vanadium Interlayer"

_metals, doi:10.3390/met11091492_

Round 1
Reviewer 1 Report
The authors worked on the nanoindentation creep behavior of Pulsed-TIG welded Inconel 718 and commercially pure titanium using vanadium interlayer. This work did a good job, and it can be considered for publication. Here are some issues which should be improved or answered prior to the acceptance for publication, such as: 1. In Materials and Methods part, the schematic of TIG welding process can be more concise, but it should be supplemented to highlight the assembly of the welding samples. 2. The details of all Figures can be handled better, such as scale, annotation and typesetting. 3. In Results and Discussions part, the chemical composition of phases (TiNi, Ti2Ni, Ni3V, etc.) should be given by EDS results. In other words, the table of chemical composition of phases marked in Fig. 2 should be supplemented. In addition, it is suggested to delete Fig. 3. 4. The Conclusions part should be carefully summarized and improved. 5. The unit of the full text should carefully check to ensure that it is correct and consistent. 6. The English of the paper should be checked carefully and polished.Author Response
please see the attached file

Reviewer 2 Report
This manuscript presented a nanoindentation creep behavior Investion on a dissimilar metal welded joint. the microstructural characterization of this joint was first conducted with SEM with EDS. and then the hardness, and ambient temperature creep were investigated by nanoindentation. relavent works has the potential to optimize welding process of dissimilar metal welded joint.
the English language of the paper is OK, the general quality of the paper is good for that journal,and i think it can be published in the journal of Metals after minor revision. below are some comemts:
1) Figure 6. The XRD spectrum: the title of the X axis changes to: “2Ï´ (degree)”
2) Figure 10. Hardness/Elastic Modulus vs. V to IN718 plots: How many times these tests have been repeated? Why the standard deviations of each point have not been added to the plots? (Idem Figures 11-14)
3) the manuscript reffes to elastic modulus in Section 2.3.1, but didn't calculate it in the subsequent research,which is confusiong.
4) the indentaiton creep tests are conducted in the ambient temperature. However, metal creep often take place in high temperature. do them have any similarity and difference ? Please explain it.
Reviewer 3 Report
This document explores the nanohardness profiles across different weld regions and the nanoindentation creep of TIG welded joints. The authors found that the creep mechanism at room temperature is diffusional creep in which the dislocation glide dominates the creep deformation. The manuscript appears to be of good quality, but there some work is needed (see below) to improve it.
1. Scale bars should be added to the Figures 4(b)-(e) and Figures 5(b)-(c).
2. In Figure 7, please provide an inset to the figure (or make a separate figure) which highlights the pop-in behavior in Figure 7(b).
3. Did the authors perform an analysis on the pop-in behavior to characterize the behavior?
4. For the indentations from Figures 10-14, the authors should complete 5 rows of indents to improve the statistics of Figures 10(b)-14(b).
5. For the creep curves, please provide good-of-fit for the fitted equations.
6. Please provide linear fits to the curves for Figures 18(a) and 21(a) with corresponding R^2 values.
Round 2
Reviewer 3 Report
The authors did a great job at addressing my comments and therefore the paper is acceptable for publication.